# Crude Extracts of *Talaromyces* Strains (Ascomycota) Affect Honey Bee (*Apis mellifera*) Resistance to Chronic Bee Paralysis Virus

**DOI:** 10.3390/v15020343

**Published:** 2023-01-25

**Authors:** Katerina Vocadlova, Benjamin Lamp, Karel Benes, Vladimir Matha, Kwang-Zin Lee, Andreas Vilcinskas

**Affiliations:** 1Branch Bioresources, Fraunhofer Institute for Molecular Biology and Applied Ecology (IME), Ohlebergsweg 12, 35392 Giessen, Germany; 2Institute of Virology, Faculty of Veterinary Medicine, Justus-Liebig-University Giessen, Schubertstrasse 81, 35392 Giessen, Germany; 3OncoRa s.r.o., Nemanicka 2722, 37001 Ceske Budejovice, Czech Republic; 4Retorta s.r.o., Tresnova 316, 37382 Borsov nad Vltavou, Czech Republic; 5Institute for Insect Biotechnology, Justus-Liebig-University Giessen, Heinrich-Buff-Ring 26-32, 35392 Giessen, Germany

**Keywords:** *Apis mellifera*, antiviral activity, CBPV, fungal extracts, *Talaromyces*, mycotoxins

## Abstract

Viruses contribute significantly to the global decline of honey bee populations. One way to limit the impact of such viruses is the introduction of natural antiviral compounds from fungi as a component of honey bee diets. Therefore, we examined the effect of crude organic extracts from seven strains of the fungal genus *Talaromyces* in honey bee diets under laboratory conditions. The strains were isolated from bee bread prepared by honey bees infected with chronic bee paralysis virus (CBPV). The antiviral effect of the extracts was also quantified in vitro using mammalian cells as a model system. We found that three extracts (from strains B13, B18 and B30) mitigated CBPV infections and increased the survival rate of bees, whereas other extracts had no effect (B11 and B49) or were independently toxic (B69 and B195). Extract B18 inhibited the replication of feline calicivirus and feline coronavirus (FCoV) in mammalian cells, whereas extracts B18 and B195 reduced the infectivity of FCoV by ~90% and 99%, respectively. Our results show that nonpathogenic fungi (and their products in food stores) offer an underexplored source of compounds that promote disease resistance in honey bees.

## 1. Introduction

The pollination of plants by insects is a key ecosystem service in agricultural and natural habitats [1]. An estimated 88% of flowering plants are pollinated by animals [2], and 35% of agricultural production is dependent on this process [1]. The recent decline in pollinator activity has therefore raised concerns about the potential impact on biodiversity, the economy and the quality of human life [3,4].

The western honey bee (*Apis mellifera* L., 1758) (Hymenoptera: Apidae) is a generalist pollinator of many plant species [1,4,5,6]. Multiple factors affect colony health and survival, including the availability of nutrients, beekeeping practices, and the spread of parasites and pathogens [4,7]. Honey bees can be infected by bacteria, fungi, protozoa, and viruses [7,8,9,10]. Until recently, viral diseases were considered a low risk, despite occasional outbreaks of infections caused by acute bee paralysis virus (ABPV), chronic bee paralysis virus (CBPV), black queen cell virus (BQCV), and sacbrood virus (SBV). However, the prevalence and virulence of some viruses has increased significantly in the last two decades and is linked to the transfer of the parasitic mite *Varroa destructor* from its original host–the Asian honey bee (*Apis ceranae* Fabricius, 1793)–to *A. mellifera* [11,12,13,14]. These mites harm honey bees by feeding on their hemolymph and fat body, thus impairing their immunity, reproduction and vigor, and, consequently, reducing the likelihood of overwinter survival [14,15]. *V. destructor* is also responsible for the transmission of deformed wing virus (DWV) and viruses representing the so-called AKI complex (ABPV, Kashmir bee virus and Israeli acute paralysis virus) [16,17,18].

A significant deterioration in bee health has been observed in recent decades, which is most evident in the poor overwintering results recorded in the northern hemisphere. Annual surveys reported average winter losses of 43.7–44.8% in the USA [19,20,21,22] and up to 20.9% in Europe from 2012 to 2017 [23,24,25]. Honey bee losses cannot be attributed to a single cause, but mites and viruses are important factors, and effective control measures are therefore urgently required [26]. Current best practices to control viral diseases in honey bees include introduction of bee stocks with hygienic traits and the management of *V. destructor* infestation using acaricidal treatments [27]. Although control measures against mites are effective, it is not possible to eradicate them completely due to their complex life cycle, which leads to annually recurring problems in beekeeping [27,28]. The control and treatment of viral infections in honey bees has been investigated in many studies [29,30], but it has thus far been impossible to break the cycle of recurring mite infestations and viral infections in order to establish a biological balance [27,31,32,33].

Natural products are screened for novel compounds against viruses [34] that can also have potential application in beekeeping. Recent studies have shown that some phytochemicals can protect honey bees against parasites and viruses, including caffeine and thymol [35,36,37]. Furthermore, extracts of some polypore fungi have been shown to inhibit honey bee viruses under laboratory and field conditions [38]. Nonpathogenic filamentous fungi are found naturally in honey bee hives, for example in corbicular pollen and bee bread, and their main function is presumably to preserve the pollen [39,40,41]. Such commensal fungi have also been shown to antagonize the entomopathogenic fungus *Ascosphaera apis* (Maassen ex Claussen, L.S. Olive and Spiltoir, 1955), which causes chalkbrood disease [42,43], and to regulate the microbial community in stingless bees [44]. However, the role of the fungal community in honey bee hives has not been studied in detail.

We isolated fungi from bee bread and identified several strains of the genus *Talaromyces* (Benjamin, 1955) (Eurotiales: Trichocomaceae). This mold is commonly found in soil, plants, and foods [45]. *Talaromyces* species have previously been identified in honey [46], dead *A. dorsata* (Fabricius, 1793) adults [47], and in the nest and food stores of stingless bees representing the genus *Mellipona* (Illiger, 1806) [48]. In industrial biotechnology, *Talaromyces* species are used to produce enzymes, pigments, and bioactive compounds, including those with antibacterial, antifungal, and antiviral activity [45,49,50]. Given the ability of *Talaromyces* species to produce antiviral compounds [49,51], we decided to test the effect of their crude organic extracts on CBPV infections in *A. mellifera*. CBPV is a taxonomically unclassified, enveloped RNA virus [52] and was identified as the causative agent of chronic bee paralysis in the 1960s [53,54,55]. Recently, the virus has gained attention due to the rapid increase in the number of cases in some parts of Europe [12]. The symptoms of the disease are easy to identify, and the infection follows classical dose-mortality kinetics under laboratory conditions [53]. CBPV is therefore an ideal model system to test the antiviral activities of fungal extracts in honey bees. Accordingly, we compared the survival of CBPV-infected bees fed on a diet enriched with fungal extracts from *Talaromyces* vs. a control diet. We also tested the antiviral activity of the extracts in mammalian cell culture experiments using model-enveloped and nonenveloped viruses. The analysis of fungal extracts with natural antiviral activity may facilitate the development of biological control methods that improve colony health and fitness and, thus, support agricultural productivity and food security. Such extracts could also be suitable as antiviral therapeutics for human use.

## 2. Materials and Methods

### 2.1. Preparation of Fungal Crude Extracts

Fungal strains were isolated from honey bee bread collected in Kamenny Malikov, Czech Republic (49°12′51.533″ N, 15°7′5.129″ E) in March/April 2019. Following their characterization, they were deposited as *Talaromyces* strains in the Fraunhofer strain collection (EXT111748–EXT111754). Seven strains were cultivated in duplicate on Sabouraud dextrose agar (SDA) for 1 week at 25 °C. The mycelia were transferred to a 100 mL malt peptone medium comprised of 30 g/L Difco malt extract (Thermo Fisher Scientific, Hessen, Germany) and 5 g/L mycological peptone from meat (Carl Roth, Karlsruhe, Germany) at pH 5.4 ± 0.2. The cultures were incubated at 28 °C for 6 h, shaking at 175 rpm (ø = 50 mm), and then at 26 °C and 70% relative humidity for a further 18 days as a static culture.

The cultures were freeze-dried and extracted with HPLC-grade methanol (VWR Chemicals, Darmstadt, Germany). Briefly, we added 40 mL of methanol and incubated the samples in an ultrasonic bath for 5–10 min before shaking them for 2 h at 175 rpm (ø = 50 mm). The samples were then passed through filter paper, and the extraction process was repeated. The crude extracts were then dried in preweighed Falcon tubes using a rotary vacuum concentrator (Christ, Frankfurt, Germany), redissolved in 80% acetone, and diluted to a final concentration of 8 mg/mL. The crude extracts and diluted stocks were stored at –20 °C.

### 2.2. Honey Bee Survival Assay

#### 2.2.1. Honey Bee Feeding and Rearing in vitro

Honey bees were collected from two colonies in the apiary of Justus-Liebig University Giessen, Germany (50°34′06.7″ N 8°40′19.7″ E), between June and August 2021. A piece of a capped brood comb containing late-stage pupae was incubated at 35 °C and 70% relative humidity. Newly emerged bees were collected every 24 h, transferred to conical plastic boxes (18 × 18 × 7.9 cm) and fed on 1:1 (w/v) sucrose solution *ad libitum*. After 3 days, groups of 18–25 bees were housed in conical experimental plastic boxes (11.5 × 8.5 × 8 cm) containing a piece of sterile wax and were fed on the same sucrose solution supplemented with either 40 µg/mL of the fungal extracts or solvent alone (80% acetone) until the end of the experiment. The feeding tubes were weighed and replaced every third day. Feed consumption was recorded as the weight difference before and after tube replacement. A control box identical to the experimental boxes without bees was placed in the incubator to measure the rate of evaporation of the sucrose solution. The average value of the evaporation control was subtracted from the final volume of the solution consumed.

#### 2.2.2. Virus Titration—End Point Replication Assay (BID_50_)

The infectious inoculum was acquired from honey bee pupae infected with CBPV by injection, as previously described [56]. After incubation for 5 days and euthanasia at –20 °C, the bees were placed individually into 2 mL screw-cap tubes containing 830 µL of phosphate-buffered saline (PBS), seven 0.5 mm zircon beads, and two 3 mm glass beads, followed by homogenization (2 × 45 s) in a FastPrep device (MP Biomedicals, Santa Ana, CA, USA). The samples were cleared by centrifugation (18,000 g, 1 min, room temperature), and the supernatants were pooled, aliquoted, and stored at –80 °C.

To determine the infectious loads, the inoculum was titrated and reinjected into honey bee pupae. Capped honey bee brood was cut out of bee combs from healthy bee colonies (specified below) and incubated at 35 °C and 70% relative humidity. Blue-eyed bee pupae (days 13–15 of development) were prepared from the combs and transferred individually to the wells of 24-well plates. Before injection, the virus suspension was thawed on ice, centrifuged (5000 g, 10 min, 4 °C), and a 10-fold dilution series up to 1:10^10^ was prepared in PBS. Eight pupae were injected in the thorax with 1 µL of each dilution or PBS as a negative control. The pupae were incubated until the time of emergence (day 21 of development). Virus infection was deduced from the death of infected pupae with developmental stagnation 2 or 3 days after inoculation. The half-maximum bee pupae infectious dose (BID_50_) of the suspension was calculated using the Spearman and Kärber algorithm (Marco Binder calculator, Department of Infectious Diseases, Molecular Virology, Heidelberg University).

#### 2.2.3. Infection of Adult Honey Bees

After 8 days of feeding with the fungal extracts, the bees were injected with CBPV to assess the effect of the extracts on the infection. To avoid repeated CO_2_ exposure, only 3–5 adult bees were anesthetized at a time (within ~10 s) before injection with 1 µL of diluted CBPV into the thorax using a Nanoject II device (Drummond, Birmingham, AL, USA). The living and/or dead bees were counted every day until the mortality reached 100%. Honey bees that died within 24 h post-injection were excluded from the analysis to ensure that only deaths caused by viral infection were counted rather than those caused by handling or wounding.

The experiment was performed in 3–6 replicates and the data were combined. Based on the mortality rates before CBPV injection, 56–124 honey bees were infected for each fungal extract, with the exception of extract B195. The dietary addition of this extract caused high mortality, so only 10 bees were injected in the pilot experiment. Due to the obvious high toxicity of extract B195, this treatment was excluded from subsequent experiments. The data were evaluated in GraphPad Prism 9 using the Kaplan–Meier survival analysis tool.

### 2.3. Antiviral Activity of the Fungal Extracts in Mammalian Cell-Based Assays

#### 2.3.1. Virus Stock Solutions

An end-point dilution assay was carried out to determine the half-maximum tissue culture infectious dose (TCID_50_) of the nonenveloped feline calicivirus (FCV strain F-9; ATTC VR-782) and the enveloped feline coronavirus (FCoV strain WSU 79-1146; ATTC VR-990) on Crandell-Rees feline kidney (CRFK) cells [57]. The virus suspensions were prepared as a 10-fold dilution series. The diluted viral suspensions (80 µL) were incubated in 96-well plates with 50 µL of the cell suspension (1 × 10^4^ CRFK cells) at 37 °C for 3 days. The TCID_50_ and viral titer were calculated using the Spearman and Kärber algorithm.

#### 2.3.2. Replication/Infection Inhibition Test with Fungal Crude Extracts

We tested four extracts that showed different effects in the bee pupae survival test (B11, B13, B18, and B195). A 10-fold serial dilution of each extract was prepared by mixing 20 µL of the extract with 180 µL of a complete cell culture medium consisting of Dulbecco’s modified Eagle’s medium (DMEM) supplemented with 10% heat-inactivated fetal calf serum (FCS). The addition of 50 µL of the cell suspension (1 × 10^4^ cells) was followed by incubation at 37 °C for 3 h. We then added 50 µL of the virus suspension (1:20 dilution, corresponding to 1 × 10^4^ TCID_50_ or ~7.5 × 10^3^ plaque-forming units (PFU), and the plate was incubated at 37 °C for 3 days. The number of viral plaques was counted using an inverted cell culture microscope.

#### 2.3.3. Virus Inactivation Test with Fungal Crude Extracts

We mixed 50 µL of the concentrated FCV and FCoV stock (2 × 10^5^ TCID_50_) with the extracts in a 1:10 ratio in 96-well plates. We used 80% acetone as a solvent control. The mixture was incubated for 3 h at 37 °C before preparing a 10-fold dilution series with the pretreated virions. For the measurement of the remaining infectious dose, we added 150 µL of the cell suspension (1 × 10^4^ cells) and incubated at 37 °C for 3 days. Viral titers were determined by the evaluation of cytopathogenic effects.

### 2.4. Analysis of Fungal Compounds

#### 2.4.1. UHPLC-HR-MS Analysis and Metabolite Annotation

We dried 200 µL of each extract (80 mg/mL) in the rotary vacuum dryer, as described above, then redissolved the residue in 40 µL of methanol and stored it at 4 °C overnight. The samples were centrifuged (8000 g, 1 min, room temperature) and 30 µL aliquots were transferred to glass vials and processed as previously described [58]. Briefly, the samples were fractionated on a 1290 UHPLC system (Agilent, Santa Clara, CA, USA) equipped with DAD, ELSD, and maXis II (Bruker, San Jose, CA, USA) ESI-qTOF-UHRMS. We used a gradient of 0.1% formic acid in water (buffer A) and 0.1% formic acid in acetonitrile (buffer B) at a flow rate of 600 µL/min. The gradient began at 95% A and was held for 0.30 min before a transition to 4.75% A over 18.00 min and 0% A over 18.10 min, with a hold for 22.50 min. The gradient then increased to 95% A over 22.60 min followed by a hold for 25.00 min. The column oven temperature was set at 45 °C, and the column was an Acquity UPLC BEH C18 1.7 µm (2.1 × 100 mm) with an Acquity UPLC BEH C18 1.7 µm VanGuard Pre-Column (2.1 × 5 mm).

Raw data were processed with DataAnalysis v4.4 (Bruker) using recalibration with sodium formate, followed by RecalculateLinespectra with a threshold of 10,000 and subsequent FindMolecularFeatures (0.5 to 25 min, S/N = 0, minimal compound length = 8 spectra, smoothing width = 2, correlation coefficient threshold = 0.7). Bucketing was performed using ProfileAnalysis v2.3 (Bruker; 30–1080 s, 100–6000 *m/z*, Advanced Bucketing with 24 s 5 ppm, no transformation, Bucketing basis = H1). Bucket annotations were performed using Metaboscape v3.0 (Bruker) based on the HR- *m/z* ratios and isotope patterns. The Bucket annotation list was manually compiled and consisted of ubiquitous *Talaromyces* sp. compounds reported in the literature (Appendix A). Fragmentation patterns of the annotated buckets were additionally manually investigated and compared (where available) to those reported in the public GNPS spectral database.

#### 2.4.2. Molecular Networking Analysis

UHPLC-QTOF-MS/MS data were analyzed by molecular networking to allow the variable dereplication of known and unknown metabolites. Firstly, the raw data (.d files) were converted to plain text files (.mgf) containing MS/MS peak lists using MSConvert (ProteoWizard package) [59], wherein each parent ion is represented by a list of fragment *m/z* value pairs (peak picking—vendor MS level = 1–2; threshold—absolute intensity, 1000 most intense). Molecular networking followed established protocols [60] using a cosine similarity cutoff of 0.7. Additionally, ions needed a minimum of six shared fragments (tolerance Δppm 0.05) with at least one partner ion to be included in the final network. In silico fragmentation predictions for a commercial database of compounds [61] (AntiBase 2017) [62] and our in-house reference compound MS/MS database were included in the network as reference substances to narrow down the molecular structure and to highlight compounds of interest. CytoScape v3.4.0 [63] was used to visualize the data as a network consisting of nodes and edges, wherein each node represents a parent ion and its color reflects the sample from which the MS/MS file was obtained. The edge width represents the cosine similarity score between nodes (thick edges indicate high similarity), and the size of the nodes represents the relative abundance of the ion in the extract.

## 3. Results

### 3.1. Survival Analysis

#### 3.1.1. Preparation of Extracts and Feeding Assays

Based on ITS sequence analysis, the seven fungal strains were identified as *Talaromyces* with the highest similarity to *T. purpureogenus* (data not shown). Characterization of the strains, including phylogenetic analysis, is still underway, so we cannot yet definitely confirm the species designation. The dried methanol extracts of the seven selected *Talaromyces* strains (Figure 1) resulted in yields of 1.1–5.5 g (Table 1).

The average daily consumption of sugar solution containing the fungal extract or acetone as a control was 19.68 mg/bee and did not differ significantly among the groups (one-way ANOVA; F = 0.63, *p* = 0.741; Figure 2).

The mortality of the bees after feeding for 8 days reached 1.67–6.67% in most of the groups, but the mortality rate was 50% in the group fed with extract B195. Accordingly, B195 was excluded from further experiments after the first group of 10 bees was tested. In the other experimental groups, 56–124 bees were injected (Table 2).

#### 3.1.2. Determination of an Effective CBPV Infection Dose

The injection of bee pupae with 1 µL of the 10^6^-fold CBPV dilution resulted in the death of all inoculated individuals, whereas the 10^7^-fold CBPV dilution killed 75% (Figure 3), and the 10^8^-fold (or higher) dilution did not differ in effect from the mock infection group injected with PBS. CBPV infection was not detected by RT-PCR in healthy emerging bees representing the 10^7^-fold dilution group. The BID_50_/mL was 3.16 × 10^10^, corresponding to a load of ~2.18 × 10^10^ PFU/mL (calculated using the factor of 0.69). Due to 100% CBPV infection, which resulted in the death of all pupae in the group injected with the 1:10^6^ dilution of the CBPV stock, we decided to use this dilution for the evaluation of the fungal extracts.

There were significant differences in the survival curves among the groups (log-rank test; χ^2^ = 258.80; *p* < 0.0001). The statistical difference between the survival of the control and CBPV-infected groups (log-rank test; χ^2^ = 86.14; *p* < 0.0001) confirmed that the experimental setup was robust. Significant differences compared to the CBPV-infected group were observed in five of the groups fed on the extracts (Figure 4). However, the observed differences in bee survival were positive in only three of the five groups: B13, B18 and B30 (Figure 4B–D).

### 3.2. Antiviral Activity of the Fungal Extracts in Mammalian Cell Culture

#### 3.2.1. Pre-Treated Cells

Virus growth was observed in CRFK cells pretreated with the dilution series of most fungal extracts 3 days after infection with 1 × 10^4^ TCID_50_ of FCV or FCoV as revealed by the near-complete destruction of the monolayers due to viral lysis. However, in the case of extract B18, we observed a strong and dose-dependent suppression of viral replication, with mostly healthy cells in higher concentrations. The 1:10 dilution of the extract (concentration = 800 µg/mL) was not cytotoxic but strongly inhibited the replication of both viruses and even resulted in the development of an intact monolayer for the cells infected with FCV (Figure 5). Extracts B11 and B195 were highly cytotoxic at a dilution of 800 µg/mL in the noninfected control cells and showed no antiviral protection at higher dilutions. Therefore, any potential antiviral effects of higher concentrations could not be investigated in this assay.

#### 3.2.2. Pretreated Virions

We also incubated the virus suspension with high concentrations of the extracts to investigate their direct virucidal effects. The active ingredients were diluted out during subsequent titration of the virus suspensions, which limited the influence of cellular factors. Contrary to our expectations, the enveloped virus FCoV was unaffected by the solvent-only treatment, which comprised a 1:20 dilution of 80% acetone for 3 h, despite the known mode of action of lipid solvents against viruses [64]. Indeed, the virus titer in the acetone-treated sample was 4 × 10^6^ TCID_50_/mL, whereas 1:20 dilutions of extracts B18 and B195 (400 µg/mL) reduced the virus titer to 1.5 × 10^5^ and 1.5 × 10^4^ focus-forming units (FFU)/mL, respectively (Figure 6). The other extracts had no effect on the FCoV titer. Again, contrary to our expectations, the solvent-only treatment resulted in the complete inactivation of FCV, which is a nonenveloped virus. Due to the unexpectedly strong influence of the solvent, we were unable to evaluate the virucidal effect of the extracts against FCV.

### 3.3. Metabolite Annotation

We were able to annotate Rubratoxins A and B, with an annotation confidence level of two as defined by the Chemical Analysis Working Group of the Metabolomics Standards Initiative [65], on the basis of their molecular formulae and identical fragmentation patterns with those deposited in the public GNPS spectral database. The molecular networking analysis revealed the presence of several derivatives (Appendix A). An in-depth analysis of these derivatives was out of the scope of the present report.

## 4. Discussion

Numerous studies have focused on the biotic and abiotic factors that influence the success of overwintering in honey bees [7,66,67]. However, the colonization of the hive by microbes, including nonpathogenic fungi, has received little attention thus far.

Bee bread contains fungi that are acquired either from honey bee salivary secretions or the collected pollen. Some of these fungal genera are repeatedly found in bee bread [39,40]. However, to the best of our knowledge, this is the first report describing the genus *Talaromyces* isolated from the bee bread of *A. mellifera*. *Talaromyces* has been found in honey [46], in dead *A. dorsata* (Fabricius, 1793) adults [47], and in stingless bees (*Melipona* spp.) [48]. Multiple studies have reported a relationship between *Talaromyces* spp. and insects from different orders over the last decades [68]. We conducted the first study on the antiviral effects of organic extracts of different *Talaromyces* strains isolated from bee bread.

Our in vivo experiments showed that extracts of *Talaromyces* strains isolated from bee bread had diverse biological effects. The addition of three extracts improved the survival rate and prolonged the average lifespan of CBPV-infected honey bees, but others had no significant impact or were toxic to the bees and therefore increased the mortality rate. Such experiments are complex because they merge direct and indirect effects that are difficult to disentangle, so we complemented the in vivo assay with an in vitro system by testing the fungal extracts in an established mammalian cell culture model. We exposed treated and untreated CRFK cells to the enveloped virus FCoV and the nonenveloped virus FCV. The most potent extracts derived from strains B18 and B195 repressed viral replication in a dose-dependent manner. Although the cells were preincubated with the fungal extracts and only subsequently infected with the viruses, we cannot completely rule out the possibility that the effects were due to virucidal properties of compounds in the extracts. A second virus inactivation assay showed that extract B18 indeed inactivated ~99% of FCoV virions. However, it is likely that additional pharmacological mechanisms were involved in the suppression of viral replication because the virucidal effect alone could not explain the low effective dose of the extract. Several antiviral compounds have been isolated from *Talaromyces* spp., including vanitaracin A, coculnol, and chrodrimanin F [51,69]. The in-depth analysis of compounds in our extracts was beyond the scope of the current study but will be investigated in the future.

The toxic effects of some extracts in honey bees were also confirmed in the cell culture model, where the high cytotoxicity of these extracts prevented the determination of antiviral properties. We assume that such toxic effects are associated with mycotoxins, such as rubratoxins, produced by some *Talaromyces* strains. Recently, rubratoxin B, produced by *T. purpureogenus*, was shown to have insecticidal activity against locusts [70]. Honey bees are exposed to many types of mycotoxins in pollen, such as aflatoxins and ochratoxins, fumosins, zearalenone, and deoxynivalenol, produced mainly by fungi of the genera *Aspergillus*, *Penicillium* and *Fusarium* [71]. Some *Talaromyces* strains used in our study produced rubratoxins A and B (B11, B30, B49, B69 and B195). However, no rubratoxins were found in the extracts of strains B13 and B18, which had a positive effect on honey bee survival. These data suggest that the presence of rubratoxins can increase the mortality of honey bees, although extract B30 (also containing rubratoxins) promoted honey bee survival, so the interactions appear to be more complex. The concentrations of these mycotoxins in the extracts, and the concentrations that are tolerable for honey bees, remain unknown and will be investigated in future experiments.

Optimal conditions for fungal growth and mycotoxin production can deviate significantly. Such differences are also observed among fungal species and between different strains of the same species. Mycotoxin production is often strain-dependent, which is consistent with our results. Furthermore, fungal growth and mycotoxin production depend on many other factors, such as temperature, water availability, pH, and illumination [72,73].

To make pollen storable and to increase the digestibility of its components, honey bees mix collected pollen with nectar and salivary gland secretions and keep it in their wax cells to let it ripen, generating so-called bee bread. The mixture undergoes complex fermentation processes, mainly influenced by lactic acid bacteria and various fungi [39]. The acidic pH, low water activity, high oxidation-reduction potential, and presence of competitive microorganisms create unfavorable conditions for microbial growth, thus protecting bee bread from spoilage [41]. However, some microbes, especially molds and yeasts, can grow slowly in such an environment. Some *Talaromyces* species are particularly tolerant of dry conditions and can grow at a water activity of 0.82, whereas other fungi can grow in a water activity range of 0.61–0.85 [46]. Furthermore, the presence of nutrients and the absence of competitive bacteria under these conditions can broaden the range of temperatures and water activities suitable for spore germination and growth [72], allowing the production of short (<10 µm) hyphae by a few germinating spores [39]. In the dormant state, the metabolism of fungal spores is shut down to approximately 50% [39,74], but metabolic activity increases greatly during the first phase of germination, where swelling occurs prior to germ tube formation [75].

Our results provide the first description of *Talaromyces* strains in hives of the honey bee (*A. mellifera*) and the relationship between fungal bioactivity and honey bee health under laboratory conditions. By producing antiviral compounds and mycotoxins, *Talaromyces* can protect bees from viruses but can also cause detrimental effects. A more realistic simulation of natural conditions would help us to understand the balance between toxic and antiviral compounds in bee bread colonized by *Talaromyces* and the relevance to honey bee antiviral resistance. This would also be facilitated by the identification of specific compounds that confer the antiviral and toxic effects.

## Figures and Tables

**Figure 1 viruses-15-00343-f001:**
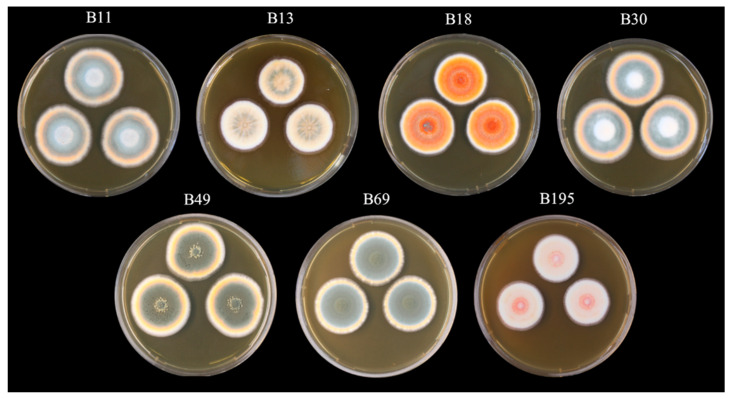
Seven strains of *Talaromyces* used in this study. The cultures were grown on malt extract agar at 25 °C for 1 week.

**Figure 2 viruses-15-00343-f002:**
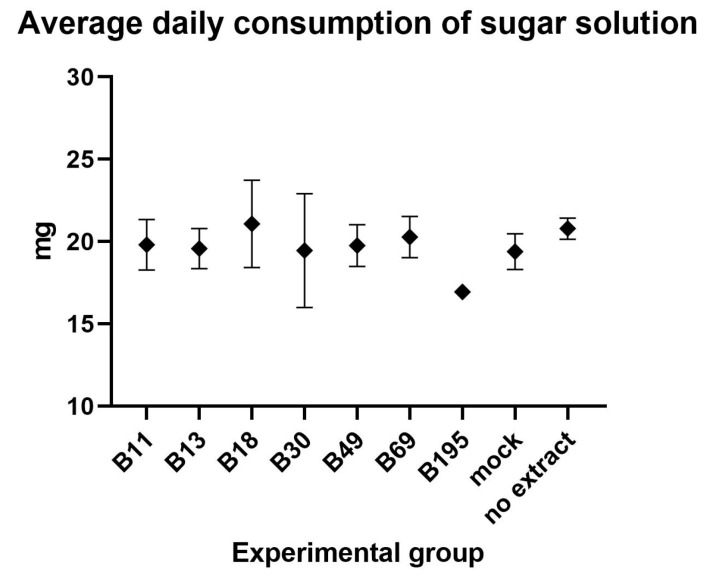
The average daily consumption (mg/bee) of 1:1 (w/v) of sugar solution enriched with the fungal crude extracts, the extraction solvent (80% acetone) in the mock group, or no extract, before the injection of CBPV.

**Figure 3 viruses-15-00343-f003:**
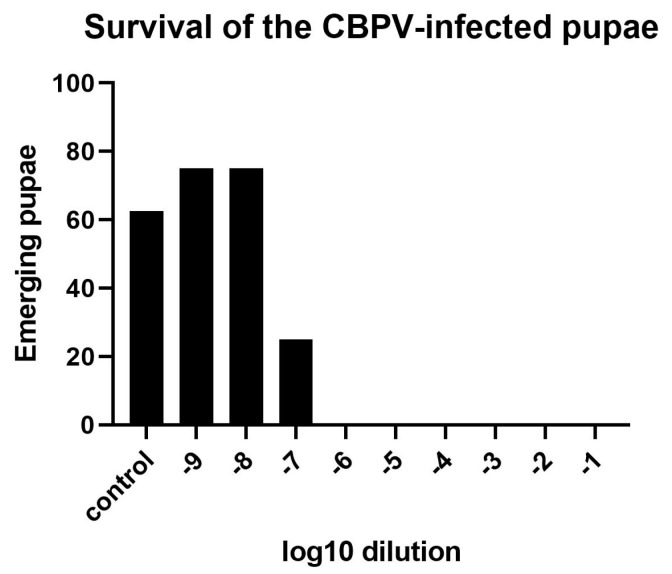
The percentage of surviving (emerging) pupae injected with serial dilutions of the CBPV viral titer and PBS (control).

**Figure 4 viruses-15-00343-f004:**
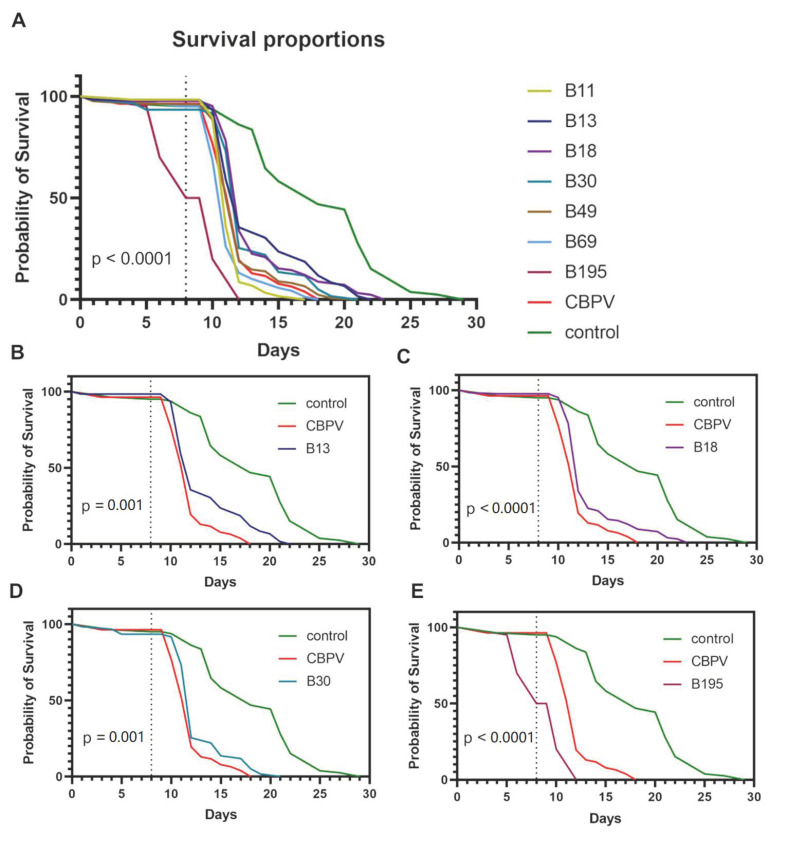
Survival proportions of bees fed on sugar solution containing fungal extracts or solvent (80% acetone) followed by the injection of CBPV or mock infection (control). The dotted line denotes the day of CBPV injection. The top graph shows the results for all seven extracts (**A**). For clarity, the extracts with significant effects are also shown in separate graphs (**B**–**E**), with the exception of B69, which was only just below the significance threshold (*p* = 0.044).

**Figure 5 viruses-15-00343-f005:**
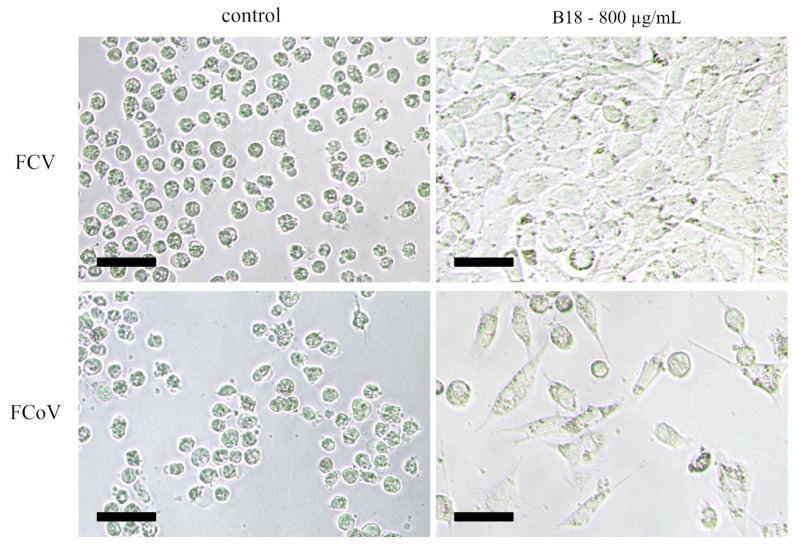
CRFK cells infected with FCV (top left) and FCoV (bottom left) in the control (solvent only) wells. The addition of the B18 extract inhibited the replication of both viruses (right) with a stronger effect on FCV. Note the healthy and infected cells after FCoV infection (bottom right) and the healthy monolayer after FCV infection (top right). Scale bar: 50 µm.

**Figure 6 viruses-15-00343-f006:**
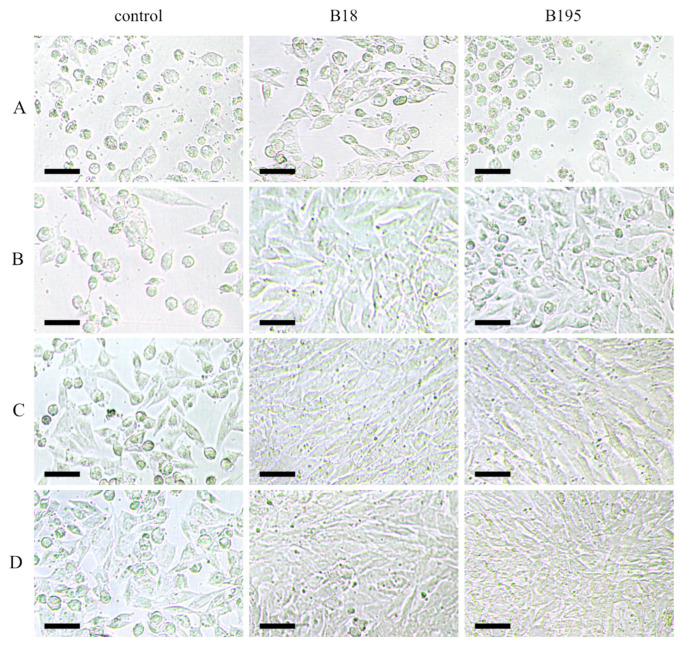
The FCoV stock of 2 × 10^5^ TCID_50_ (control (**A**–**D**)) caused visible symptoms of infection in the CRFK cells, as revealed by the strong cytopathic effects. Pre-incubation of extracts B18 and B195 with the FCoV stock reduced the viral titers by ~99% and ~90%, respectively. Note the cytotoxic effects of the extracts at higher concentrations (B18 and B195, row (**A**)), and the absence of infection in rows (**B**–**D**). Rows (**A**–**D**) represent the 10-fold dilution series of the virus stocks treated with the solvent (control) or extracts B18 and B195. Scale bar: 50 µm.

**Table 1 viruses-15-00343-t001:** Crude methanol extract yields from the seven *Talaromyces* strains used in this study.

Fungal Strain	Crude Extract Yield [g]
B11	5.52
B13	3.47
B18	1.05
B30	4.42
B49	5.33
B69	3.54
B195	1.24

**Table 2 viruses-15-00343-t002:** The mortality of bees fed on sugar solution enriched with fungal crude extracts before the CBPV injection or the solvent (80% acetone) in the mock group (followed by mock infection) or no extract in the no extract group (followed by CBPV injection).

Experimental Group	Mortality [%]	No. of Injected Bees
B11	1.67	59
B13	1.67	59
B18	2.36	124
B30	6.67	56
B49	4.00	120
B69	5.00	76
B195	50.00	10
no extract	3.75	77
mock acetone	5.00	76

## Data Availability

Not applicable.

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
