# Peer review of "Crude Extracts of Talaromyces Strains (Ascomycota) Affect Honey Bee (Apis mellifera) Resistance to Chronic Bee Paralysis Virus"

_viruses, 2023, doi:10.3390/v15020343_

Round 1

Reviewer 1 Report

Honeybee is a generalist pollinator which is important for maintaining the ecological balance and agricultural production. However, the number o honeybee has being continuously decreased in the past. Honeybee viruses were always one of the main factor threatening honeybee healthy. Current hygienic beekeeping and the management of V. destructor infestation using acaricidal treatments could not effectively solve the damage caused by virus. The effective virus control measures are still urgently required. In the present study, the authors explored the possibility of using the extracts of fungi from bee bread to inhibit honey bee viruses CBPV. They found that some of the extracts could mitigated CBPV infections and increased the survival rate of bees. And the test in mammalian cells further certificated the virus inhibiting action. The research is practical valuable in beekeeping. Here are the three suggestions.

1.       Figure 2. add a title for the vertical axis such as “daily consumption of sugar solution”

2.       In Figure 3, only the survival data of control group has the variation data. Theoretically, -9, -8 and -7 dilution groups should also have the variation data. Please supply them in the Figure.

3.       Line 283: the “Figure 1” here should be revised as the Figure 4. Moreover, the separate graphs in the Figure should be labeled with A, B, C, D etc.

4.       It suggested to combine the core content of Table 3 with the survival curves of all seven extracts in Figure 4, to make the content more concise.

Author Response

The authors would like to thank the reviewer for helpful comments and suggestions. We appreciate your time and insight. We imcorporated changes reflecting all the suggestion.

Figure 2. add a title for the vertical axis such as “daily consumption of sugar solution”

Response: Thank you for this important point. The title was probably cut by accident during the edition process. The title has been added (page 6, line 270).

2. In Figure 3, only the survival data of control group has the variation data. Theoretically, -9, -8 and -7 dilution groups should also have the variation data. Please supply them in the Figure.

Response: We agree, this is confusing. Since there were only 4 individuals injected in 2 repetitions (only to determine the optimal CBPV infection dose), we only show the percentage of surviving pupae without variation data (page 7, line 292).

3. Line 283: the “Figure 1” here should be revised as the Figure 4. Moreover, the separate graphs in the Figure should be labeled with A, B, C, D etc.

Response: We appreciate this note and we edited the figure according to the suggestion (page 8, lines 300-308).

4. It suggested to combine the core content of Table 3 with the survival curves of all seven extracts in Figure 4, to make the content more concise.

Response: Thank you for the suggestion. We implemented it to make the content more concise (page 8, line 303).

Reviewer 2 Report

The manuscript describes a study on a really current topic. Such study has good potential for high citation. I recommend to accept this manuscript after minor revisons which I did in enclosed PDF file. My congratulations to this interesting study.
